# Molecular Mechanism of Induction of Bone Growth by the C-Type Natriuretic Peptide

**DOI:** 10.3390/ijms23115916

**Published:** 2022-05-25

**Authors:** Estera Rintz, Grzegorz Węgrzyn, Toshihito Fujii, Shunji Tomatsu

**Affiliations:** 1Department of Molecular Biology, Faculty of Biology, University of Gdansk, 80-308 Gdansk, Poland; estera.rintz@ug.edu.pl (E.R.); grzegorz.wegrzyn@ug.edu.pl (G.W.); 2Skeletal Dysplasia Research Lab, Nemours Children’s Health, 1600 Rockland Rd., Wilmington, DE 19899, USA; 3Department of Diabetes, Endocrinology, and Nutrition, Kyoto University Hospital, Kyoto 606-8507, Japan; tfujii@kuhp.kyoto-u.ac.jp; 4Department of Pediatrics, Thomas Jefferson University, Philadelphia, PA 19107, USA

**Keywords:** natriuretic peptide, growth plate, chondrocyte cell, bone growth, molecular mechanism

## Abstract

The skeletal development process in the body occurs through sequential cellular and molecular processes called endochondral ossification. Endochondral ossification occurs in the growth plate where chondrocytes differentiate from resting, proliferative, hypertrophic to calcified zones. Natriuretic peptides (NPTs) are peptide hormones with multiple functions, including regulation of blood pressure, water-mineral balance, and many metabolic processes. NPTs secreted from the heart activate different tissues and organs, working in a paracrine or autocrine manner. One of the natriuretic peptides, C-type natriuretic peptide-, induces bone growth through several mechanisms. This review will summarize the knowledge, including the newest discoveries, of the mechanism of CNP activation in bone growth.

## 1. Introduction

The skeletal development process in the body occurs through sequential cellular and molecular processes called endochondral ossification. Only craniofacial bones and clavicles are not developed through endochondral ossification [1]. Endochondral ossification occurs in the growth plate where chondrocytes differentiate from resting (reserve), proliferative, hypertrophic, to calcified zones. Differentiated chondrocytes construct a cartilage extracellular matrix (ECM) through the upregulation of proteolytic enzyme production (matrix metalloproteinases). Together with the vascular endothelial growth factor, this process induces degradation of cartilage matrix components inducing vascularization. Eventually, cartilage will be replaced with mature bone tissue. Differentiation of growth plate cartilage is regulated by several factors, including hormones, growth factors, and components of the cartilage ECM [2].

Natriuretic peptides (NPTs) are peptide hormones with multiple functions, including the regulation of blood pressure, water-mineral balance, and many metabolic processes [3]. NPTs secreted from the heart (the cardiac atria and ventricles-ANP and BNP; vascular endothelium-CNP) activate different tissues and organs, working in a paracrine or autocrine manner. The first NPT was discovered as atrial NPT (ANP) in 1981, secreted from the cardiac atria and ventricles [4]. From that time, eight NPTs have been described: ANP, B-type natriuretic peptide (BNP), C-type NPT (CNP), D-type NPT (DNP), urodilatin, uroguanylin, osteocrin, myocardin, and musclin [3]. The functions of three main peptides (ANP, BNP, and CNP) were investigated in detail.

The peptides mentioned above can be used to diagnose heart or pulmonary diseases (Table 1). Various cellular factors stimulate the expression of genes coding for ANP, BNP, and CNP, and the secretion of these peptides. In the case of patients with heart failure, levels of particular forms of ANP (proANP/γ-ANP, β-ANP) or BNP (uncleaved proBNP, mature BNP, and N-terminal proBNP) increased compared to healthy controls. Precursors of the peptides are proteolytically degraded to their active forms, acting in tissues and organs [5]. Moreover, a decrease in the levels of NPTs and an increase in the receptor level that degrades the peptides (NPR-C) have been associated with diet and physical activity [6].

There are three known NPT receptors: NPT receptor-A (NPR-A; also called NPR1 or GC-A), NPT receptor-B (NPR -B; another way NPR2 or GC-B), and NPT receptor-C (NPR-C; also called NPR3 or GC-C). ANP and BNP are ligands for NPR-A, while CNP activates NPR-B. NPR-C receptor is responsible for the degradation and internalization of circulating NPTs within the cell [84]. NPRs catalyze the synthesis of cyclic guanosine-3′,5′-monophosphate (cyclic GMP; cGMP), known as the internal mediator for most effects of NPTs.

CNP structure differs from ANP and BNP in the second cysteine, lacking further C-terminal extension [85]. CNP is released by endothelial cells, cardiomyocytes, and fibroblasts and regulates the function of the cardiovascular system together with ANP and BNP [86]. However, CNP function not only affects the cardiovascular system but also governs bone growth [87,88,89,90,91]. Interestingly, the mechanism of CNP action can be different depending on the cell type. CNP activates ossification and calcification in bone cells, while in the valvular interstitial cells, those processes are inhibited by the CNP peptide [92]. The same pattern is observed with osteoblast and myofibroblast differentiation, where those processed are activated in bone cells and inhibited in the valvular interstitial cells with CNP treatment [92].

Some previous reviews explained CNP peptide and its bone mechanism, but still questions remain unanswered. This review will summarize our knowledge, including the newest discoveries, on the mechanism of CNP activation in bone growth. MeSH (Medical Subject Headings) was applied to select articles in this review process. In PubMed search, we have used the keywords of “C-Type Natriuretic Peptide bone growth”, “C-Type Natriuretic Peptide”, “CNP bone”, and “C-Type Natriuretic Peptide mechanism bone”. A total of approximately 40 articles were found under the CNP activating bone mechanism. We have also searched the clinical trials via https://clinicaltrials.gov/ (accessed on 4 April 2022) to prepare Table 1.

## 2. C-Type Natriuretic Peptide and Nitric Oxide as Bone Growth Regulators

CNP as a bone growth regulator was first mentioned in 1995 [93], where CNP was found to induce bone resorption. The production of CNP can be induced by cytokines (such as IL-1α, IL-1β, and TNFα), thus affecting osteoclast resorption. It is unclear if CNP acts directly on osteoclasts or not. When CNP was added to the culture, cGMP production increased (as well as bone resorption), meaning that the receptor for CNP (NPR-B) is present in the bone marrow culture cells. Adding the CNP inhibitor to the culture decreased the efficiency of bone marrow resorption in CNP-treated and untreated cells, indicating the bone marrow released CNP. Nevertheless, it is unclear what kind of cells secreted CNP peptide, as bone marrow is a heterogeneous mixture of the cells [93]. Not only resorption of osteoclasts but also proliferation and differentiation of osteoblasts are regulated by CNP peptides. Production of cGMP, induced by CNP, leads to bone formation in osteoblast-like cells. CNP activation increased the activities of alkaline phosphatase (ALP) and osteocalcin and the mineralization of nodules in the cell culture [94]. Similarly, CNP upregulated bone turnover biomarkers, both osteoblastic (osteocalcin, procollagen type I, total ALP, and bone-specific alkaline phosphatase) and osteolytic (tartrate-resistant acid phosphatase-TRAP and cross-linked C-terminal telopeptide of type I collagen- CTX-I) in osteoblast-like cells [91].

Another molecule in bone growth regulation is nitric oxide (NO). Mice deficient in the eNOS show bone abnormalities, while inducible NOS (iNOS) deficient mice show imbalances in bone osteogenesis and abnormalities in bone healing [95,96,97]. iNOS is synthesized in response to inflammatory stimulation, producing NO, thereby inducing bone loss. As NO is a highly reactive molecule, molecular bone targets are not fully understood [98]. Nevertheless, NO inhibits bone resorption by inducing cGMP synthesis (the same as CNP) [99]. Thus, CNP can affect bone remodeling [93].

Research on the interactions of NO and NPTs has shown that only ANP can inhibit NO through the NPR-A receptor in macrophages [100]. Even though all natriuretic receptors (with two types of NPR-B receptor) are produced in macrophages, CNP still does not inhibit NO. Receptor for CNP exists in two NPR-B receptors that differ in structure [101], while CNP is a ligand for both. However, one of the forms is shorter, without the ability to induce cGMP production by CNP binding [100]. NO is synthesized from L-arginine by inducible nitric-oxide synthase (iNOS) [102]. CNP neither inhibits the induction of iNOS nor induces L-arginine transport, which is critical for NO production [103].

Both NO and CNP stimulate the production of cGMP in cells. An additional product of the cGMP forming reaction could be 8-nitro-cGMP (Figure 1). The level of 8-nitro-cGMP increases in the presence of CNP in bone culture in vitro. Like cGMP, 8-nitro-cGMP stimulates the proliferation of cells in the growth plate as the bone grows. These results show that NO and CNP activation mechanisms are more complicated, activating cGMP and other molecules, which may work synergistically in the bone growth effect [101].

## 3. CNP/NPR-B/cGMP Pathway

The signaling pathway of the chondroprotective effects with CNP is mediated by the NPR-B receptor, enhancing the activation and synthesis of cGMP. As a result, PKGII is activated, leading to the elevation of ECM (collagen II, aggrecan) and bone mineralization (ALP), finally inducing bone growth. Additionally, the activation of the NPR-B receptor inhibits the MEK/ERK signaling pathway, stopping cartilage degradation (Figure 1). However, this pathway can be stimulated by other factors (not only CNP) that either enhance or reduce bone growth. Aging and inflammation are some processes affecting reduced bone growth [104]. Thus, skeletal dysplasia should be treated at an early age of patients. There are also different CNP/NPR-B/cGMP signaling pathway inhibitors. One of the first described inhibitors was lysophosphatidic acid (LPA), which causes a decrease in cGMP concentration that was not correlated with the receptor loss. While previous research showed that PKC could be involved in the dephosphorylation of the NPR-B receptor at the Ser-523 residue [105], in this case, the activity of PKC was not changed. According to the authors, this response increases the response of fibroblasts to the wound-healing process [106]. Those investigations showed that CNP activation and different agents could regulate the NPR-B receptor. cGMP is converted from guanosine triphosphate in reactions catalyzed by various enzymes (guanyl, guanylyl, or guanylate cyclases). Guanylate cyclases are activated by different factors such as guanylin, uroguanylin, guanylyl cyclase-activating proteins, NO, as well as NPTs. Activation of the second intracellular messenger (cGMP) stimulates or modulates different pathways within the cell, depending on what receptor was activated (NPR-A, -B, or C). Processes that are activated by cGMP include platelet aggregation, blood pressure neurotransmission, sexual arousal, gut peristalsis, long bone growth, intestinal fluid secretion, lipolysis, phototransduction, cardiac hypertrophy, and even oocyte maturation [107].

Guanylyl cyclases are ubiquitous enzymes with two classes occurring in mammals: soluble cytoplasmic guanylyl cyclase and single membrane-spanning forms (NPR-/GC-A, -B, -C, -D, -E, -F, and -G) that can be produced in different tissues. The localization within the body of these two guanylyl cyclases classes is different, as well as their ligands. ANP and BNP ligands activate NPR-A, produced in lung, kidney, adrenal, vascular smooth muscle, endothelium, heart, and adipose. Inactivation of NPR-A leads to heart-related problems (hypertension, cardiac fibrosis, and hypertrophy). The NPR-B, activated by CNP, is produced in bones, vascular smooth muscle, lung, brain, heart, liver, uterus, and follicles [108]. ATP increases the enzyme activity of NPR-B [109]. The degradation receptor for the NPTs is synthesized in the intestinal epithelium, stimulated by bacterial heat-stable enterotoxin, guanylin, and uroguanylin. Impaired production of GC-C leads to increased proliferation of colonic epithelial cells without a change in blood pressure [107].

## 4. CNP/NPR-B/cGMP/MAPK Pathway

The CNP/NPR-B/cGMP pathway activates endochondral ossification and induces bone growth in different models. FGFs are soluble or cell surface proteoglycans (family of 22 members [110] that activate FGFR-3 tyrosine receptors on the subphase of chondrocytes [111]. Activating mutations of FGFR3 are associated with skeletal diseases like achondroplasia, hypochondroplasia, or thanatophoric dysplasia [112]. Signal transduction of the FGFR-3 phosphorylates ERK1/2, a protein kinase of the MAPK signaling pathway [113]. Constantly phosphorylated ERK1/2 is associated with achondroplasia [114]. On the other side, the FGFs family is associated with slowing long bone growth. Intracellular interactions of FGFs/FGFR-3/MAPK and CNP/CG B/cGMP pathways were studied in mouse chondrogenic cell lines [115]. CNP signaling activation of cGMP was reduced by FGF-2 and FGF-18, while expression of the cGMP itself was not FGFs-dependent (Figure 1). CNP and cGMP inhibited the activation of pERK1/2 by FGFs/FGFR-3. CNP did not affect STAT-1 phosphorylation by FGFs/FGFR-3, another signaling pathway that could be activated (Figure 1). CNP treatment of fetal mouse tibias induced longitudinal bone growth, increasing in size and the number of chondrocytes. This effect was attenuated when FGF18 was added to the culture [115]. Crosstalk was visible between those two pathways, with evidence of the CNP influence on bone growth.

Further research confirmed the interaction of CNP signaling pathway inhibition of FGF-2 by both direct and indirect mechanisms [116]. CNP inhibited phosphorylation and activation of ERK protein kinase induced by the FGF-2 pathway in the rat chondrocyte model. CNP blocks the ERK mitogen pathway at the level of Raf-1, but not Ras or FRS2. To inhibit Raf-1, protein kinase G (PGK) is required because the inhibited CNP pathway was not able to block the ERK/MAPK pathway. The FGF-2 pathway without CNP induces degradation of chondrocyte ECM in a rat cell model of chondrosarcoma. With CNP, FGF-2 was partially inhibited, leading to the expression, activation, and release of ECM molecules and proteins (including matrix metalloproteinase 2 (MMP2), -3, -9, -10, and MMP13). Moreover, CNP induced matrix-mediated production independently of FGF-2 [116] (Figure 2).

None of the previous studies show the direct influence of CNP on the FGF-23 expression. FGF-23 negatively regulates the expression of a gene coding for collagen-X [117] and disturbs collagen X metabolism [118]. Collagen X is produced primarily on the hypertrophic zone in the growth plate of cartilage, although it is involved in calcifying the cartilage [119]. Collagen II and aggrecan are primarily produced in the proliferative zone. One of the most recent reports describing studies on the primary rat osteoblast-like cells showed that CNP stimulates osteoblast proliferation with increased collagen-X levels. While CNP downregulated FGF-23 production, its receptor FGFR-1/Klotho was not changed. These results suggest that CNP can stimulate collagen X production through FGF-23 inhibition [91].

## 5. CNP/NPR-B/cGMP/pCREB Pathway

Mutations in either the NPPC gene encoding CNP [88] or the NPR2 gene coding for the CNP receptor called NPR-B [120] result in marked dwarfism with skeletal defects. Further studies confirmed acromesomelic dysplasia Maroteaux-type syndrome (MIM #602875) with a mutation in NPR2 [121]. Very recent studies on the Maroteaux syndrome revealed a different mechanism caused by the mutations in the NPR2 gene. Three different mutations p.Leu314Arg), p.Arg371*, and p.Arg1032* lead to variable phenotypes in the patients. These mutations affect the activation and function of the guanylate cyclase: binding of the CNP (ligand) to the NPR-B homodimer; depletion of the three domains in the receptor (transmembrane, protein kinase, and guanylate cyclase); and the deterioration of the guanylate cyclase domain. Those results indicated that the critical function of the NPR-B receptor is the activation of the cGMP in the cell [122].

Interestingly, in epiphyseal chondrodysplasia, Miura type (ECDM; OMIM #615923) disease with the heterozygous NPR2 pathogenic variant caused opposite symptoms to those in the homozygous mutation. So far, only four families with this disease have been reported. These activating mutations in NPR2 cause tall stature, long bones, scoliosis, and elongated toes and fingers in patients. The heterozygous variant (c.2647G>A (p.Val883Met)) is characterized by amino acid alternation localized in one of the domains in the NPR-B receptor-carboxyl-terminal guanylyl cyclase domain [123].

A mouse model with a gain of function mutation in the NPR2 gene exhibited skeletal overgrowth with a thick hypertrophic zone in the cartilage growth plate (higher level of the collagen type X than WT) [124]. The same pattern was not visible in the proliferating zone (similar collagen type II and aggrecan levels to WT mice). Moreover, there was no difference in apoptotic cells in the growth plate when comparing WT with Npr2 mice. Those results suggest that the NPRB receptor regulates chondrocyte maturation at later stages of development in the hypertrophic layer. It was possible to screen for molecular pathway activation using a cell model for endochondral ossification (ATDC5). Cells were cultured for several weeks (0–8) and treated with CNP to activate the NPRB receptor. Several genes were checked to determine at which stage the chondrocytes are. The main transcription factors critical for chondrocyte gene expression are Runx2, Sox9, and MEF2C [125]. Sox9 is a transcription factor involved in chondrocyte differentiation [126], Sox9 was increased after two weeks of culturing, then declined in untreated cells once cells reached the mature stage (hypertrophic cells) [124]. On the other hand, Runx2 is expressed during chondrocyte hypertrophy [127], did not decline after two weeks, and stayed at the same level. CNP treatment did not affect the expression of both genes, Sox9 and Runx2 [124], while genes encoding CNP and its receptor NPRB peaked at two weeks. The expression level of the cell proliferation gene Ccnd1, encoding Cyclin D1 responsible for the progression of the cell cycle in the G1-phase [128], was the highest at four weeks of chondrogenic induction and decreased after that time [124]. When treated with CNP, a similar effect was obtained even if CNP was added at a different time (six or eight weeks), upregulating the expression of Ccnd1. Bone growth is regulated by the parathyroid hormone that is present at the end of long bones and delays chondrocytes hypotrophy [129], thus the expression of the genes coding for a parathyroid hormone-related protein (Pthrp) and its receptor parathyroid hormone 1 receptor (Pth1r) were measured in cell culture. Expression of both Col2a1 and Pthrp was upregulated after CNP treatment, as was Pth1r even six weeks after treatment with CNP. Moreover, NPR-B was active when CNP was added to the ATDC5 cells due to the markedly increased production of cGMP [124].

Treatment of hypertrophic chondrocytes with cell-permeable cGMP resulted in increased phosphorylation of CREB, the transcription factor stimulating the production of the Cyclin D1, thus, showing chondrocyte activation through the NPRB-mediated molecular pathway [124]. Those experiments were also confirmed in cartilage isolated from WT neonates and treated with CNP (increased phosphorylation of CREB and Cyclin D1 expression). Contrary to previous research [130], ERK1/2 was not phosphorylated in the primary chondrocytes [124]. In the FGFR-MAPK pathway, FGFR3 is produced in the proliferating and pre-hypertrophic zone of the growth plate, not in the hypertrophic zone [131]. Thus, the CNP pathway can activate and regulate chondrocytes in different ways.

The expression of the Npr3 gene, encoding the NPR-C receptor, was elevated after CNP treatment, which is consistent with previous research with the organ culture [132]. In the primary chondrocyte cell culture, Sox9, Prhrp, and Ihh (Indian hedgehog) expressions were increased after CNP treatment, suggesting CNP involvement in both proliferation and differentiation of the cells. Moreover, when cells were treated with CNP, cell numbers increased. With the addition of the inhibitor for either CREB or cyclin D1, CNP-induced NPRB activation was abolished, suggesting the apparent involvement of CREB and cyclin D1 in skeletal overgrowth [124]. The action of parathyroid hormone and its receptor is mediated through cAMP/protein kinase A (PKA)/CREB, regulating chondrocytes by keeping them in the proliferation stage [129]. Nevertheless, CNP treatment increased cGMP production in the primary chondrocytes, with no cAMP effect. CNP induced phosphorylation of CREB even with PKA inhibitor treatment while it suppressed the phosphorylation of CREB by PTHrP [129]. The same pattern of CREB phosphorylation through NPRB independent of PKA was demonstrated recently [124].

Similarly, when protein kinase-G II (PKGII), downregulated by CNP, is removed from cartilage, cartilage matrix production and chondrocyte proliferation were disturbed in mice [133]. The overexpression of CNP and activating mutations in the Npr2 gene positively affects cartilage matrix production and chondrocyte proliferation. In a previous experiment, a systemic knockout of the CNP gene was made. The generation of cartilage-specific knockout (using Cre recombinase (Cre)-loxP) of the genes revealed differences in endochondral bone growth in the growth plate between CNP and its receptor [88]. Knockout in the NPR-B receptor resulted in shorter mice than CNP knockout mice. There are several explanations for why it happened. First, and most likely, secretion of CNP from other than cartilage tissues (like blood vessels) causes its binding to the receptor, resulting in lower impairment of growth. Second, there is a higher ligand (CNP) amount than its receptor (NPR-B). When the receptor is missing and the ligand is still present, the general effect on endochondral growth could be more substantial if there is a knockout in the receptor rather than the ligand itself. Furthermore, the sensitivity of the NPR-B receptor could increase due to the absence of CNP. Additionally, the clearance system of the peptides could not work correctly because of the downregulation of the NPR-C receptor [134] or an increased level of osteocrin, a natural ligand for the NPR-C receptor [135]. Due to osteocrin’s attachment to the NPR-C receptor, the clearance system of the NPTs is not working. If NPTs cannot be cleared, they will be ligands to receptors that they generally do not activate. The NPR-B receptor may be activated by other ligands, such as ANP or BNP (normally reacting with the NPR-A receptor) [88]. In both models, hypertrophic chondrocytes in the growth plate were drastically reduced with chondrocytes proliferation [88].

While inhibition of the MEK1/2-ERK1/2 pathway stimulates bone growth, p38 MAPK is one of the mediators in the CNP signaling pathway [136]. p38 inhibits stimulation of endochondral bone growth and delays mineralization in tibia organ culture by CNP. These studies confirmed that CNP targets hypertrophic chondrocytes in the growth plate rather than other zones. Even though the CNP receptor is present in all three zones at a similar level, CNP signaling pathway kinases (PKG-I and -II) are primarily produced in the hypertrophic cells. This can potentially explain the spatial distribution of CNP effects. Moreover, CNP induces the expression of NPR2 and NPR3, suggesting the existence of a feedback loop to limit CNP signaling. CNP also regulates bone morphogenic proteins (BMP) signaling and cell adhesion processes [136].

## 6. C-Type Natriuretic Peptide as Extracellular Matrix Regulator

CNP is engaged in the intracellular processes required for bone growth and the extracellular matrix (ECM) [137]. CNP increases the number of chondrogenic condensations in the micro mass culture of mouse embryonic limb cells. Production of N-cadherin, a cell adhesion molecule, increased after treatment with CNP peptide. Glycosaminoglycan synthesis (chondroitin sulfate) was also stimulated, probably through increased levels of the enzyme (xylosyltransferases I) necessary to produce cartilage glycosaminoglycans. On the other hand, chondrogenic transcription factors (Sox9, -5, and -6), modulators of ECM, as well as ECM building proteins (collagen II and aggrecan) were not changed by CNP [137]. Another study investigated CNP and the formation of the nodules with biglycan (BGN), a small proteoglycan involved with bone formation in osteoblasts. BGN mRNA expression increased with CNP, which shows CNP’s ability to regulate gene expression to be closely correlated with bone mineralization [138].

## 7. Synergic Effect of OP-1 and CNP

Mutation in the gene coding for the growth hormone-insulin like growth factor-I (IGF-1) leads to prenatal and postnatal growth failure. Interaction of growth hormone with IGF-1 regulates endochondral bone growth. IGF-1 modulates bone growth through the paracrine mechanism rather than systematically. The synergic mechanism of osteoblast differentiation and proliferation by IGF-1 and osteogenic protein-1 (OP-1) was observed when both agents were added to the rat osteoblastic cell culture [139].

Bone morphogenetic proteins (BMPs) are growth factors that induce bone growth, and the main downstream molecules for BMPs are Smads. Activated Smad5 plays an important role in BMP target gene activation. After phosphorylation, Smad5 interacts with Smad4, making a complex translocated to the nucleus, activating BMP target genes [140]. OP-1 activated Smad5 through phosphorylation of the C-terminal region.

Similarly, the synergic mechanism of OP-1 and CNP was observed in the phosphorylation of the Smad5 transcription factor responsible for bone morphogenic protein gene activation [141]. While CNP has been added alone to the primarily osteoblastic cells, activation of Smad5 was not observed. Alkaline phosphatase (ALP) induction by OP-1 was enhanced by CNP, leading to the mineralization of bone nodules. CNP+OP-1 also enhanced the protein expression level of bone morphogenic protein receptors IB and II (BMPR-IB; -II), compared to OP-1 alone. CNP induces the expression of genes involved in the bone mineralization process. Nevertheless, the Runx2 protein level was not enhanced by CNP but was elevated after treatment with OP-1 alone, and it was slightly reduced by the combination treatment [141].

## 8. Survival of Animals with C-Type Natriuretic Peptide

The survival rate of the mice with total CNP knockout was around 40% [142], which was due to severe malocclusion [88]. Mice with the CNP knockout could not eat by themselves; however, when food was pulverized, the survival rate increased to ~70% [88]. The cartilage specific CNP knockout mice survival rate was almost the same as total CNP knockout mice (75.0%). The survival rate of about ~70% is caused by their impaired skeletal growth [88]. As the mice could have higher mortality, it was impossible to analyze them in adult age. For this reason, a generation of a rat model with CNP knockout was performed. The phenotype of the rats was similar to that of the mouse model, manifesting short stature with endochondral bone growth retardation. Compared to the mice model, rats survived for over one year without malocclusions visible in the mouse model. Interestingly, only bones were affected by CNP deficiency in histological analysis, and no other tissues studied were changed compared to WT littermates. Growth plate width in the bone of CNP knockout rats (mainly hypertrophic chondrocyte layer) was lower together with reduced proliferation of growth plate chondrocytes [143]. On the other hand, crossing animals with circulating CNP peptide (SAP-Nppc-Th mice) with chondrodysplastic CNP-depleted mice leads to increased survival. Moreover, circulation of CNP recovers chondrodysplastic CNP-depleted animals from impaired endochondral bone growth, together with increased body weight [87].

## 9. C-Type Natriuretic Peptide as a Treatment

Achondroplasia is one of the most common diseases leading to dwarfism (1:25,000) by the mutation in a single gene FGFR3 coding for fibroblast growth factor receptor 3). In 80% of the cases, de novo dominant mutation occurs in the egg or sperm that forms the embryo [144]. Parents with new mutations usually do not have a growth deficit. The gene mutation probability increases if the father is more than 35 years old. It is the most common disease in the group of growth defects diseases. Symptoms include dwarfism, midfacial hypoplasia, incorrect torso length, excessive lordosis of the lumbar spine, narrowing of the spinal stricture, small cubic vertebral bodies, prominent forehead, collapsed nose bridge, micromelia (small hands), with normal intellectual development [112]. FGFR3 inhibits long bone growth through different pathways. Activation of phosphorylation of STAT-1 regulated by FGFR3 inhibited chondrocyte proliferation. Moreover, in the MAPK pathway regulated by FGFR3, phosphorylation of ERK1/2 inhibited chondrocyte differentiation. Overall, activation of FGFR3 seems to inhibit the proliferation and differentiation of chondrocytes [130].

The mechanism of CNP in achondroplasia shows that CNP inhibit the MAPK pathway, through decreasing FGF-2 activated level of pERK1/2, what results in animals’ normal length [145]. However, CNP did not affect the level of phosphoSTAT-1 in the cartilage of neonatal mice [145]. The achondroplasia mouse model, with FGFR3 mutation leading to jaw deformities, was crossed with mouse overexpressing CNP [146]. Fgfr3ach/SAP-Nppc-Tg mice had significantly improved the craniofacial region comparing symptoms to the Fgfr3ach mouse model. The thickness of synchondrosis was noticed, with increased chondrocytes proliferation within the craniofacial cartilage. Moreover, enhanced endochondral bone growth in the combined mouse model ameliorated the foramen magnum stenosis compared to Fgfr3ach only mice [146]. CNP induces endochondral ossification in the long bones and helps restore craniofacial bones in the achondroplasia mouse model.

Achondroplasia is the most common growth retardation disorder. However, growth retardation can also be a side effect of glucocorticoid-based drug administration [147]. Recent studies showed that high doses of glucocorticoid inhaled by children with asthma correlate with growth retardation [148]. The mechanism of glucocorticoids directly act on chondrocytes within the growth plate, including proliferation and differentiation, together inhibiting bone growth [147]. Growth hormone therapy to restore growth retardation connected to glucocorticoid-based drugs is ineffective [149]. Exogenous CNP-53 is a stable molecular form of intrinsic CNP peptide. It was administered to mice with induced glucocorticoid drug growth retardation [147]. Glucocorticoid-based drug (dexamethasone; DEX) and CNP-53 (dose 0.5 mg/kg/day) were injected into four- to eight-week-old mice by subcutaneous injection. The effect of CNP depended on age, as it was more effective in four-week-old mice. The molecular mechanism of this effect is not based on the action of DEX on cGMP level induction by CNP but instead on the reduction of Erk1/2 phosphorylation by combined treatment [147].

In another study, four-week-old CNP-knockout (CNP-KO) and WT rats were subcutaneously infused with 0.15 or 0.5 mg/kg CNP-53 peptide daily for four weeks. There was no significant difference in plasma bone turnover markers between WT and CNP-KO rats. The short stature of the CNP-knockout mice was restored with circulating CNP peptide administration, where CNP-KO rats were more sensitive to the treatment than WT rats [150]. In acceptance with previous research [101], gene enrichment analysis using fetal rat tibia confirmed that CNP negatively influences the MAPK pathway in chondrocytes [150].

An essential part of mechanism regulation is the feedback loop. In the case of CNP, it remains unclear how it is regulated in the systemic feedback regulation. Several factors indicate that CNP can be regulated in a systemic feedback manner.

Administration of exogenous CNP (CNP-53) to four-week-old rats (0.5 mg/kg/day) for three consecutive days did not alert the blood level of CNP [151]. However, this administration decreased mRNA expression in cartilage tissues. Interestingly, female rats were more sensitive to CNP administration, with a greater decrease in mRNA level. The influence of exogenous administration on endogenous CNP levels in tibia culture and chondrogenic cell lines did not show direct autoregulation. Thus, cartilage tissue is not directly regulated by CNP administration [151].

## 10. Discussion

CNP is a paracrine growth factor widely expressed in various tissues [152]. Diverse functions include the regulation of endochondral bone growth. The most critical role in humans is recognized as the hormone in skeletal growth related to growth plate proliferation. Studies in mice show that the local production of CNP within the growth plate determines physiological endochondral bone growth [88]. The dynamic role of CNP in bone growth remains a challenge due to the rapid clearance of CNP and low concentrations in the blood. An inactive portion of the synthesized product in the growth plate (amino-terminal propeptide CNP; NTproCNP) is not subject to clearance or rapid degradation. NTproCNP is an equimolar product of CNP biosynthesis, which is easily measured in plasma. Its level in blood correlates to linear growth velocity throughout growth in both children and mouse models [153]. CNP and NTproCNP levels for both sexes were high in infancy, lower in early childhood, rising during puberty, and then falling to low adult levels. Peak levels of NTproCNP are coincident with the age of peak height velocity. Thus, CNP synthesis (as measured by NTproCNP levels in plasma) is closely related to linear growth in healthy children at all ages, resulting in an excellent biomarker of linear growth [154].

CNP products in plasma were elevated in skeletal dysplasia with profound short stature due to a disruption of the CNP receptor (NPR-B) (acromesomelic dysplasia type Maroteux), activation of the MEK/ERK MAPK pathway inhibiting NPR-B signaling, activating mutations of *FGFR3* (thanatophoric dysplasia, achondroplasia, and hypochondroplasia) [1]. Two reports support those activating mutations of NPR-B cause skeletal overgrowth [155,156], reducing plasma NTproCNP. Little is known about the factors that regulate CNP expression and translation; the details of this feedback loop require further study.

The relation between the MEK/ERK MAPK and CNP/cGMP pathways was investigated in chondrogenic cells and organ culture. Phosphorylated MEK1/2 and/or ERK1/2 inhibit cGMP generation by NPR-B. Meanwhile, NPR-B-generated cGMP inhibits MEK/ERK activation by inhibiting RAF1. Thus, overactivation of the MEK/ERK MAPK pathway leads to resistance to CNP [115]. The finding of elevated CNP levels in a population with severe short stature suggests that these individuals may also have resistance to CNP.

It is noteworthy that patients with achondroplasia affecting the CNP pathway activity have elevated plasma NTproCNP concentrations, while intracellular CNP activity is reduced [116,145,157]. In achondroplasia, the normal reciprocal antagonism between FGFR3 (inhibiting endochondral bone growth) and CNP signaling (stimulating bone growth) is disregarded by a gain of function mutation in FGR3 [115]. CNP production is subject to feedback regulation. Endogenous CNP remains subject to regulation during the administration of growth-promoting doses of the CNP analog in achondroplasia. With the restraining impact of the genetic mutation on CNP signaling, baseline levels of CNP products in plasma are elevated. During periods of accelerating long bone growth induced by the CNP analog, endogenous CNP is reduced in keeping with indirect negative feedback. Direct inhibition of CNP by exogenous CNP occurs in proportion to the overall level of hormone resistance. Unusual high values of plasma NTproCNP in adolescent years with achondroplasia patients receiving the CNP analog are unexplained and require future study [115].

Take-home message;

CNP produced in the growth plate is a potent positive regulator of linear growth.Reduced intracellular CNP pathway activity may increase CNP production; a negative feedback loop regulates CNPIn a group of skeletal dysplasias, elevated plasma levels of NTproCNP indicate the presence of tissue resistance to CNP.Long-term CNP therapy for achondroplasia and other skeletal disorders remains unknown.The interaction between the CNP/NPR-B pathway and other pathways should be explored to elucidate the bone growth.

## 11. Conclusions

When organs were cultured, several mechanisms were found to be stimulated by CNP, including chondrocyte proliferation, cell hypertrophy, cartilage matrix production, and ECM activation. When the only CNP stimulation (cGMP) product was added to the culture, the same stimulation of longitudinal growth and glycosaminoglycan synthesis was observed. Nevertheless, cGMP only could not induce the proliferation of chondrocytes without an effect on cell hypertrophy. The second messenger for CNP (cGMP) could reproduce some of the CNP effects but not all, indicating that CNP activation is a more complex mechanism [158]. The signaling pathway of the CNP peptide has not been fully understood. Thus, this review has discussed known mechanisms at present, as summarized in Figure 1 and Figure 2.

## Figures and Tables

**Figure 1 ijms-23-05916-f001:**
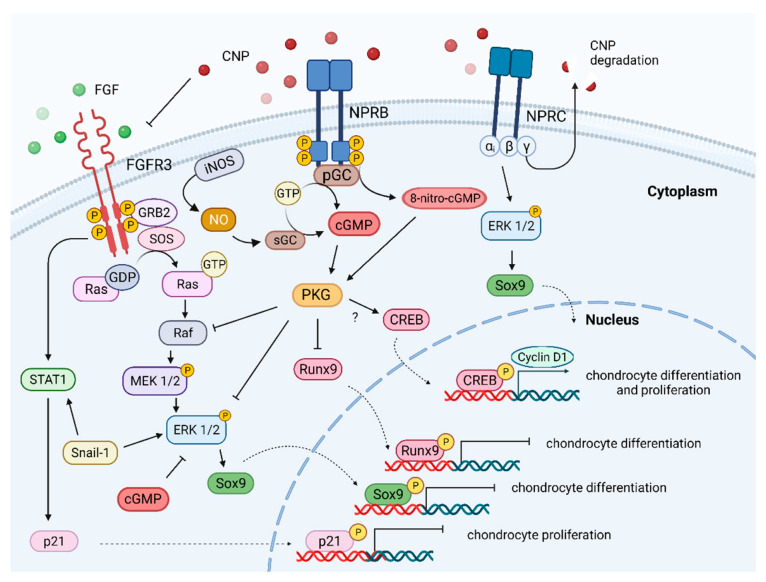
Signaling pathways activated or inhibited by CNP in the bone growth plate region. The cellular effects of CNP action are mediated through the NPR-B receptor, stimulating intracellular molecule cGMP production, which further results in the protein kinase G (PKG) activation. PGK activation by CNP results in the MAPK signaling pathway, which results in chondrocyte proliferation and differentiation. NPR-C receptor degrades CNP; additionally, Gi protein of the receptor modulates the level of ERK1/2 inside the cell. ?- it is not confirmed if CREB is activated by PKG.

**Figure 2 ijms-23-05916-f002:**
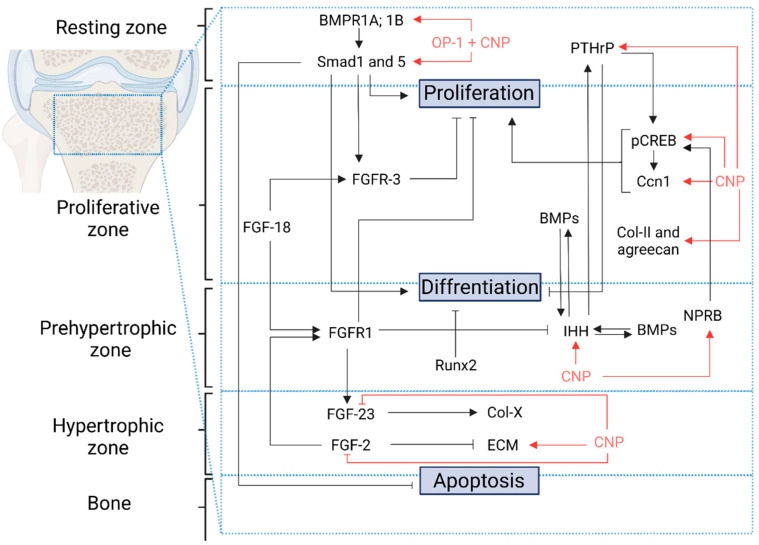
CNP and related bone growth involved protein expression patterns in the growth plate. BMPR1A; 1B: Bone morphogenetic protein receptors; Smad1; 5: Suppressor of Mothers against Decapentaplegic; PTHrP: Parathyroid hormone-related peptide; pCREB: phosphorylated cAMP-response element-binding protein; Ccn1: Cyclin 1 gene; Col-II: Collagen type 2; BMPs: Bone morphogenetic proteins; IHH: Indian Hedgehog; NPRB: NTP Receptor Type B; FGF: Fibroblast growth factor; FGFR: Fibroblast growth factor receptor; Runx2: Runt-related transcription factor 2; Col-X: Collagen type X; ECM: Extracellular Matrix Proteins; CNP: C-type natriuretic peptide. Red arrows and lines means accordingly activation or inhibition by CNP.

**Table 1 ijms-23-05916-t001:** Natriuretic peptides used in diagnosis and treatment.

NTPs	NTPs Used for Diagnosis/Evaluation		NTPs Tested for Diseases as a Treatment
	Final Peptide	Pro-form of the Peptide	
ANP	Heart failure [7,8,9,10] Cancer [11] Type-2 diabetes [12] Chronic renal failure [13]	Heart failure [10,14] Insulin resistance [15,16] Obesity [17]	Heart failure [18,19,20,21,22] Cardioprotective effects during surgery [23,24,25,26,27,28,29,30] Renal failure [31,32,33,34,35,36] Colorectal cancer [37] Forearm vasculature [37] Acute kidney injury [38] Lipid mobilization [39] Pulmonary vasorelaxant [40]
BNP	Heart failure [10,41,42,43,44,45,46,47,48,49,50,51,52,53] Acute dyspnea [54] Asthma [55] Type 2 diabetes [56] Cancer [11] Chronic renal failure [13] Obesity [17,57] Chronic kidney disease [58,59] SARS-CoV-2 [60] Atrial fibrillation [61] Hemodialisys patients [62] Diabetic nephropathy [63]	Heart failure [48,64,65,66,67,68,69] Type 2 diabetes [70,71] Obesity [17,72] Chronic kidney disease [58] Cardiac function [73] Stroke [73]	Forearm vasculature [37] Pulmonary vasorelaxant [40] Myocardial infarction [74,75]
CNP	Heart failure [47,76]	Heart failure [64] Growth failure [77]	Heart failure (CNP fused to DNP) [78] Renal failure [79,80] Achondroplasia [81,82] Anxiety [83]

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
