# Peer review of "Molecular Mechanism of Induction of Bone Growth by the C-Type Natriuretic Peptide"

_ijms, 2022, doi:10.3390/ijms23115916_

Round 1

Reviewer 1 Report

C-type natriuretic peptide function not only affects the cardiovascular system but also governs bone growth. The author reviews from multiple aspects the newest discoveries on the mechanism of CNP activation in bone growth. It is a good topic.  However, in my opinion, from the "introduction" to the” 5. CNP/NPR-B/cGMP/pCREB pathway” part, the content of the manuscript is too redundant and too messy. the manuscript mentioned a lot of bone growth-related molecules and signaling pathways involved in CNP,  the authors try to cover everything, but in fact, the readers will feel complications.

Comment:

Suggest the authors condense the parts from “introduction” to “CNP/NPR-B/cGMP/pCREB pathway” to make the article simpler. Deleted some content not involved in CNP and bone, in order to be more focused on the CNP, bone growth, and bone metabolism.

Mini comment:

Line 42-43 mentioned “eight NPTs have been described:...” , could to add the eighth one?

Line 57-58: “while CNP activates NPR-C” should be NPR-B

Line 112: understood

Line 309: “will act as ligads to” will be ... ligands to

Author Response

C-type natriuretic peptide function not only affects the cardiovascular system but also governs bone growth. The author reviews from multiple aspects the newest discoveries on the mechanism of CNP activation in bone growth. It is a good topic.  However, in my opinion, from the "introduction" to the” 5. CNP/NPR-B/cGMP/pCREB pathway” part, the content of the manuscript is too redundant and too messy. the manuscript mentioned a lot of bone growth-related molecules and signaling pathways involved in CNP,  the authors try to cover everything, but in fact, the readers will feel complications.

Thank you for your positive comments.

Comment:

Suggest the authors condense the parts from “introduction” to “CNP/NPR-B/cGMP/pCREB pathway” to make the article simpler. Deleted some content not involved in CNP and bone, in order to be more focused on the CNP, bone growth, and bone metabolism.

We delated some parts that were not CNP/bone related.

Mini comment:

Line 42-43 mentioned “eight NPTs have been described:...” , could to add the eighth one?

Thank you for your comment. In accordance with reviewer’s comment, we have added the eighth one.

Line 57-58: “while CNP activates NPR-C” should be NPR-B

Thank you for your comment. We corrected it

Line 112: understood

Thank you for your comment. We corrected it

Line 309: “will act as ligads to” will be ... ligands to

Thank you for your comment. We corrected it

Reviewer 2 Report

Dear all

I realize that authors have many journals to consider when they want to publish their work, so I appreciate your interest in IJMS; I am very happy  to be able to write in a positive way. It is evident that you have put a great deal of effort into this project and I want to praise your efforts, Fortunately, the actual contribution from your study is clear and, the manuscript as currently written suggests that it might be suitable for sharing information about this interesting and fascinating topic, but the paper that you reported, needs few minor edits. I should like to thank you for give me an opportunity to consider this work for publication. It may be that the you would like to consider resubmitting it, in which case I hope that the comments from my review may help you to revise it before resubmitting it. These comments are given below. Best Regards

  • Introduction section: references are missing in the many sentence;
  • Methods: being a revision, the materials and methods should be explained, even if it is a narrative revision, therefore I suggest to change the type of study in masterclass or add details of methods section; 
  • discussion section: I recommend adding a discussion section where you can insert some reflections of the authors and Focus on take-home messages

Author Response

I realize that authors have many journals to consider when they want to publish their work, so I appreciate your interest in IJMS; I am very happy  to be able to write in a positive way. It is evident that you have put a great deal of effort into this project and I want to praise your efforts, Fortunately, the actual contribution from your study is clear and, the manuscript as currently written suggests that it might be suitable for sharing information about this interesting and fascinating topic, but the paper that you reported, needs few minor edits. I should like to thank you for give me an opportunity to consider this work for publication. It may be that the you would like to consider resubmitting it, in which case I hope that the comments from my review may help you to revise it before resubmitting it. These comments are given below. Best Regards

We appreciate the critical and kind comments from the reviewer.

  • Introduction section: references are missing in the many sentence;

Thank you for giving us the opportunity to revise the manuscript. In accordance with

reviewer’s comment, we have added the references.

  • Methods: being a revision, the materials and methods should be explained, even if it is a narrative revision, therefore I suggest to change the type of study in masterclass or add details of methods section; 

Thank you for your suggestion. We added methods after introduction part.

  • discussion section: I recommend adding a discussion section where you can insert some reflections of the authors and Focus on take-home messages

Thank you for your suggestion. We added discussion part according to reviewer’s comment.